# SEPS: Semantic-Enhanced Patch Slimming Framework for Fine-Grained Cross-Modal Alignment

**Xinyu Mao** [1]  **Junsi Li** [1]  **Haoji Zhang** [1]  **Yu Liang** [1]  **Ming Sun** [1]

## Abstract

Fine-grained cross-modal alignment is pivotal for multimodal reasoning yet remains limited by Semantic Sparsity Bias—a fundamental asymmetry where dense visual signals are under-represented by sparse textual captions. This disparity leads to the inadvertent suppression of contextually vital visual regions stemming from patch redundancy and hinders precise concept grounding due to patch ambiguity. While Multimodal Large Language Models (MLLMs) offer rich descriptive capabilities, their naive integration often induces semantic drift due to inconsistencies with sparse ground-truth captions. To systematically resolve these challenges, we present the Semantic-Enhanced Patch Slimming (SEPS) framework. Central to SEPS is a novel Dual-Granularity Semantic Calibration mechanism, which synthesizes a Holistic Visual-Linguistic Anchor from MLLMs to synergize with original sparse queries. This mechanism transforms patch selection into a semantic consensus process, ensuring that retained patches satisfy both local discriminability and global contextual integrity. Furthermore, we propose a Salience-Guided Metric Aggregation strategy to mitigate the similarity dilution effect inherent in global mean pooling, thereby amplifying highly-relevant patch-word correspondences. Extensive experiments on Flickr30K and MS-COCO datasets demonstrate that SEPS surpasses existing state-of-the-art approaches across diverse backbones, delivering significant performance gains in text-to-image retrieval tasks. The implementation is available at https://github.com/Sweet4tars/seps.git.

[1] School of Computer Science and Engineering, University of Electronic Science and Technology of China, Chengdu, China. Correspondence to: Ming Sun <sunm@uestc.edu.cn>.

*Proceedings of the $43^{rd}$ International Conference on Machine Learning*, Seoul, South Korea. PMLR 306, 2026. Copyright 2026 by the author(s).

## 1. Introduction

Fine-grained cross-modal alignment constitutes the bedrock of modern vision-language understanding, serving as the pivotal mechanism for establishing precise correspondences between visual regions and linguistic concepts (Qian et al., 2025). This alignment capability underpins a wide spectrum of downstream applications, ranging from visual question answering (Guo et al., 2019) and image captioning (Li et al., 2019) to the increasingly demanding task of cross-modal retrieval (Fu et al., 2023). As multimodal systems evolve toward granular comprehension, the ability to accurately disentangle and align complex visual scenes with specific semantic cues has become a critical imperative (Lin et al., 2025).

Despite significant progress, current alignment paradigms universally encounter a fundamental theoretical bottleneck: **Semantic Sparsity Bias** (Wang et al., 2024; Yuan et al., 2025). This bias arises from the intrinsic asymmetry between modalities—visual inputs inherently carry dense, continuous, and high-entropy spatial information, whereas textual descriptions (captions) act as sparse, discrete, and low-entropy anchors (Liu et al., 2025a). In Vision Transformer (ViT) architectures (Dosovitskiy et al., 2020), this asymmetry manifests as two persistent challenges: *patch redundancy*, where a vast majority of visual tokens lack explicit textual supervision, and *patch ambiguity*, where the concise nature of sparse text fails to provide sufficient discriminative cues for specific visual regions. As illustrated in Figure 1(a), a generic caption such as "A woman with a tennis racket" fails to describe the surrounding context or specific visual attributes, leading to the inadvertent suppression of visually vital but textually unmentioned patches during the alignment process.

Recent advancements have sought to leverage Multimodal Large Language Models (MLLMs) to augment textual representations (Pan et al., 2023; Fu et al., 2024; Liu et al., 2025b). While promising, naive integration of MLLMs often introduces *semantic drift*, where the hallucinated or overly detailed descriptions from MLLMs conflict with the ground-truth sparse captions (Bang et al., 2025; Ravichander et al., 2025), creating noise that degrades retrieval precision. Furthermore, conventional alignment metrics typically rely

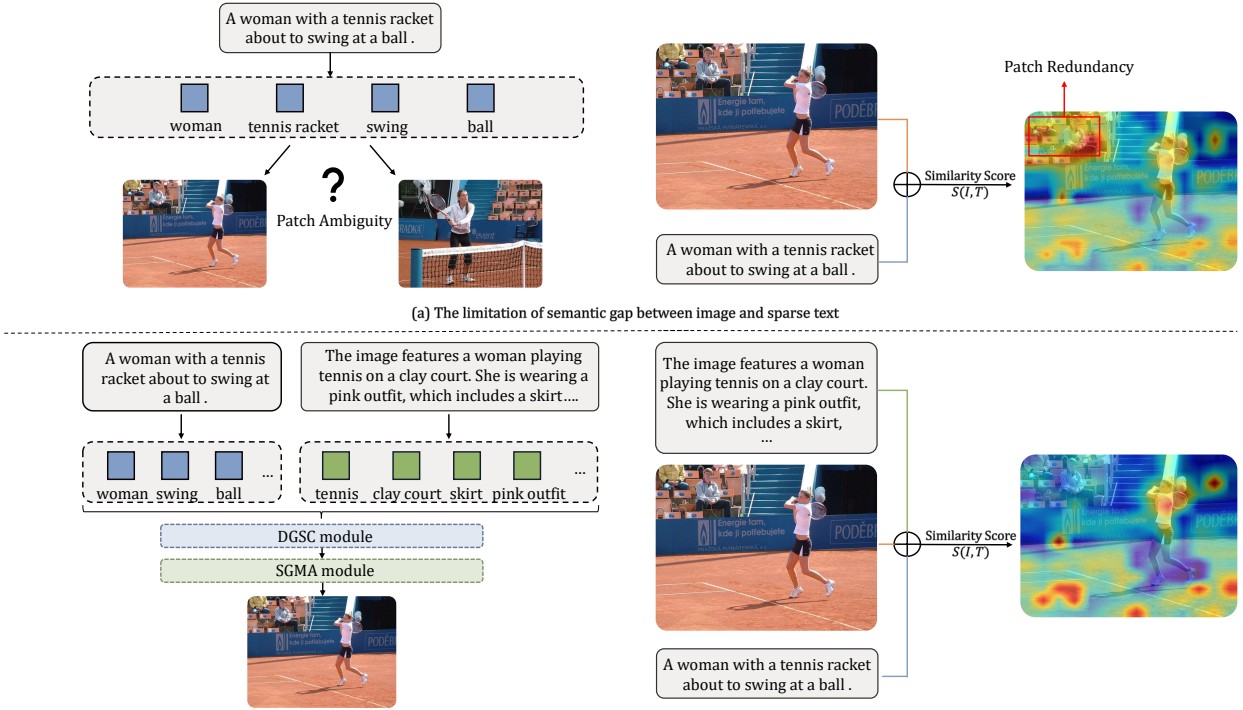

*Figure 1.* Illustration of the Semantic Sparsity Bias and the proposed SEPS framework. (a) Current works suffer from patch ambiguity and patch redundancy due to the limited semantic guidance. (b) This framework fuses semantic consensus derived from dense and sparse texts to guide visual patch selection, and introduces a salience-guided aggregation to improve patch-word alignment, which bridges the semantic gap.

on global mean pooling (Jiang et al., 2025), which indiscriminately aggregates scores from all patches. This approach is fundamentally flawed in complex scenes, as irrelevant background patches with low similarity scores effectively dilute the contribution of highly aligned foreground regions (Ge et al., 2025), obscuring the true semantic correlation.

To address these limitations, we propose the **Semantic-Enhanced Patch Slimming (SEPS)** framework, a systematic approach designed to resolve the Semantic Sparsity Bias through a novel Dual-Granularity Semantic Calibration (DGSC) mechanism. As shown in Figure 1(b), our key insight is to treat MLLM-generated content not merely as auxiliary text, but as a Holistic Visual-Linguistic Anchor. By integrating this dense anchor with the original sparse queries, we establish a semantic consensus mechanism that cross-verifies visual patches against both discriminative keywords (Sparse) and contextual narratives (Dense).

Functionally, as depicted in Figure 2(a), SEPS operates through a sophisticated pipeline. We first extract multi-view features and employ the DGSC module to identify salient visual patches. Unlike previous methods that filter patches based on single-source supervision, DGSC performs a "weighted consensus" process to preserve patches that are relevant to either the specific query or the global context. Subsequently, we introduce the Salience-Guided Metric Ag-

gregation (SGMA) module, which replaces standard mean pooling with a relevance-aware alignment strategy. This ensures that the final similarity score is dominated by the most semantically significant patch-word pairs, making the metric robust to background noise.

The principal contributions of this work are summarized as follows:

- We identify **Semantic Sparsity Bias** as the root cause of misalignment in sparse-supervised systems and propose the SEPS framework to systematically bridge the information asymmetry between dense vision and sparse language.

- We introduce the Dual-Granularity Semantic Calibration mechanism. By synthesizing unified semantics from both dense (MLLM-derived) and sparse textual modalities, this module eliminates semantic inconsistency and enables precise, context-aware patch selection.

- We develop the Salience-Guided Metric Aggregation strategy, a relevance-aware scoring paradigm that effectively mitigates the "similarity dilution" effect inherent in traditional global averaging methods.

- We demonstrate the efficacy of SEPS through extensive

experiments on Flickr30K and MS-COCO datasets, achieving state-of-the-art performance across diverse backbones, particularly in challenging text-to-image retrieval scenarios.

## 2. Related Work

### 2.1. Fine-Grained Cross-Modal Alignment

Cross-modal alignment aims to bridge the semantic chasm between vision and language, evolving primarily through two paradigms: coarse-grained and fine-grained alignment. While coarse-grained methods like VSE++ (Faghri et al., 2017) focus on global image-text similarity, fine-grained approaches seek precise correspondences between local visual features and textual tokens. Early dominant approaches relied on pre-trained object detectors (e.g., Faster R-CNN (Girshick, 2015)) to extract salient regions, followed by complex attention mechanisms for alignment (Lee et al., 2018; Diao et al., 2021; Pan et al., 2023). However, this detector-dependent paradigm suffers from high computational overhead and error propagation.

The advent of Vision Transformers (ViT) (Dosovitskiy et al., 2020) shifted the field toward grid-based patch processing. While efficient, this shift introduced intrinsic challenges: *patch redundancy* (irrelevant background) and *patch ambiguity* (lack of semantic grounding). To mitigate these, recent methods such as LAPS (Fu et al., 2024) proposed using linguistic supervision to prune redundant patches. However, these methods rely exclusively on sparse captions from standard datasets. We identify this reliance as a critical limitation: **Semantic Sparsity Bias**. Since sparse captions capture only a fraction of visual details, using them as the sole filter inadvertently discards visually rich but textually unmentioned regions. Unlike these approaches, our SEPS framework transcends single-source supervision by introducing a DGSC mechanism, integrating dense semantic contexts to rectify the biases inherent in sparse-only selection.

### 2.2. Visual-Linguistic Information Asymmetry and MLLM Integration

A fundamental bottleneck in multimodal learning is the **Information Density Mismatch**: visual signals are continuous and dense, whereas textual descriptions are discrete and sparse. This asymmetry has catalyzed the integration of MLLMs to generate dense textual descriptions as a compensatory signal.

Existing literature predominantly leverages this dense signal for representation enhancement. One line of research utilizes MLLM-generated texts to bolster long-text understanding capabilities during pre-training, exemplified by LongCLIP (Zhang et al., 2024) and LoTLIP (Wu et al.,

2024). Another stream addresses the density mismatch via asymmetric feature modeling (e.g., AVSE (Liu et al., 2025d)) or knowledge distillation, where dense text serves as a "teacher" signal to enrich sparse textual embeddings, as seen in D2S-VSE (Liu et al., 2025c).

However, a common limitation unites these works: they treat MLLM outputs merely as auxiliary data for optimizing global feature representations or textual embeddings. The potential of dense semantics to directly restructure the visual input itself remains unexplored. Diverging from these representation-centric paradigms, our work proposes a Structure-Centric approach. We utilize the MLLM-generated semantic anchor to actively guide the patch selection process, proactively filtering visual redundancy at the input level to resolve the information mismatch before alignment occurs.

## 3. Methodology

Traditional patch selection paradigms often operate under the assumption that sparse captions provide sufficient supervision for visual redundancy elimination. We challenge this assumption by identifying a critical bottleneck: Semantic Sparsity Bias. Since sparse texts ($T_s$) capture only a subset of visual details, exclusively relying on them leads to the suppression of contextually vital but unmentioned visual regions.

To resolve this Cross-Modal Information Asymmetry, we introduce the SEPS framework. Unlike preceding methods (e.g., LAPS) that perform selection based on single-granularity cues, SEPS establishes a Dual-Granularity Semantic Calibration mechanism. By constructing a "Semantic Anchor" via MLLMs, the framework enforces a consensus between discriminative keywords (sparse) and descriptive context (dense). This approach effectively bridges the semantic gap, ensuring that the selected patches are not only text-relevant but also visually self-consistent.

The framework architecture comprises three synergized modules: the Holistic Visual-Linguistic Anchor generation (Sec. 3.1), the Dual-Granularity Semantic Calibration (DGSC) module (Sec. 3.2, visualized in Fig. 2(b)), and the Salience-Guided Metric Aggregation (SGMA) module (Sec. 3.3, visualized in Fig. 2(c)).

### 3.1. Holistic Visual-Linguistic Anchor Construction

To compensate for the low information density of sparse captions, we introduce a mechanism to project visual content into a comprehensive linguistic space before alignment. We define a frozen, deterministic mapping function $\mathcal{F}_{MLLM}(\cdot)$ utilizing LLaVA (Liu et al., 2023). Given an image $I$, we generate a dense descriptive prompt $T_d = \mathcal{F}_{MLLM}(I)$.

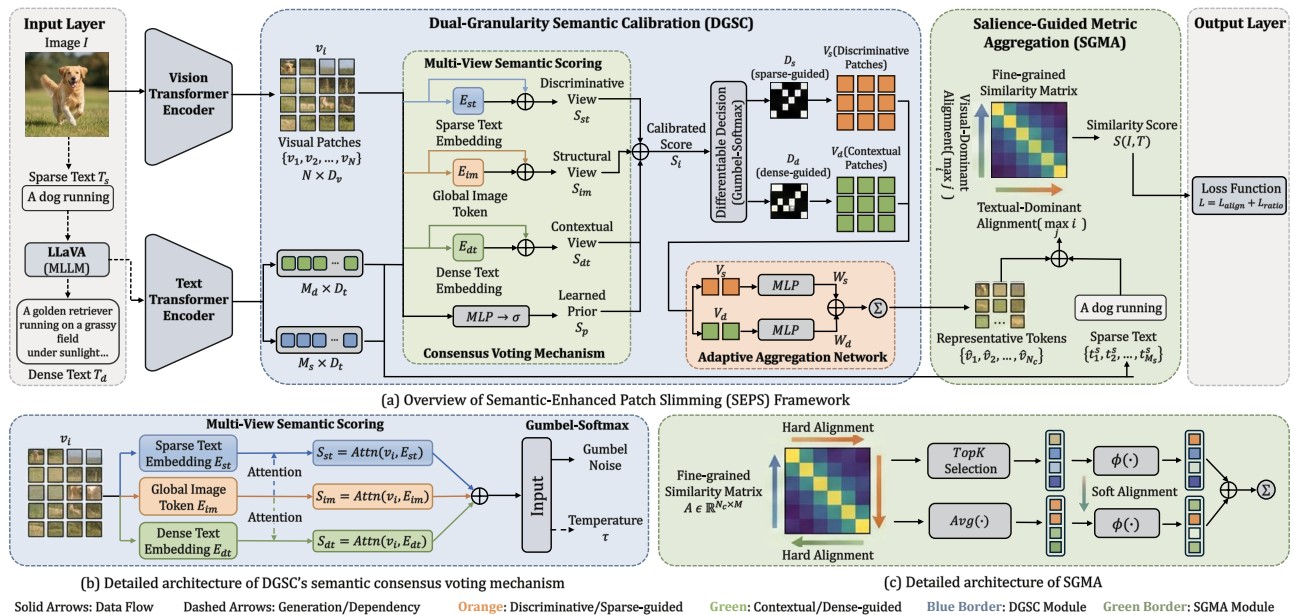

*Figure 2.* (a) Overview of the SEPS Framework for fine-grained cross-modal alignment. To mitigate information asymmetry, we first extract visual patches and textual tokens via Transformer encoders. We then introduce the DGSC module to calibrate patch importance using a semantic anchor generated by MLLMs. Finally, the SGMA module aggregates the calibrated features to compute the robust alignment score $S(I, T)$. (b)(c) Detailed architectures of the proposed calibration and aggregation modules.

Functionally, $T_d$ serves as a semantic anchor. Unlike standard textual features that only align with specific objects, this anchor encodes the global visual narrative—including background context and object relationships—into the textual modality. This ensures that the subsequent patch selection process is guided by a complete semantic blueprint ($T_s \cup T_d$), effectively neutralizing the information capacity gap inherent in traditional sparse-supervision systems.

### 3.2. Dual-Granularity Semantic Calibration

The DGSC module constitutes the core filtration engine of our framework. It transforms the patch selection task from a naive heuristic filtering process into a semantic consensus voting mechanism. By fusing unified semantics derived from dual-granularity text (Sparse & Dense), the module identifies visual patches that satisfy both local discriminability (matching specific keywords) and global contextual consistency (matching the overall narrative). This process is executed in two consecutive stages: Multi-View Semantic Scoring and Differentiable Decision with Feature Re-synthesis.

**Multi-View Semantic Scoring.** In this stage, we aim to evaluate the intrinsic informational value of each patch. We employ a score-aware prediction network to estimate a significance prior. This network, composed of a two-layer MLP with sigmoid activation, predicts a baseline score based solely on visual features:

$$s_i^p = \sigma\left(\text{MLP}\left(\boldsymbol{v}_i\right)\right), \ i \in \{1, \dots, N\}, \quad (1)$$

where $s_i^p \in [0, 1]$ denotes the learned importance prior for the $i$-th patch $v_i$.

However, visual salience does not strictly equate to semantic relevance. To bridge this gap, we construct a Tri-View Attention Manifold that aligns visual patches with three distinct semantic anchors. We argue that reliance on sparse text alone creates a "blind spot" for background context; thus, we introduce dense text relevance as a complementary view. The attention scores are computed as follows:

$$
\begin{aligned}
s_i^{st} &= \text{Norm}(v_i^T \cdot E_{st}/d) &&\triangleright \textit{Discriminative View} \\
s_i^{dt} &= \text{Norm}(v_i^T \cdot E_{dt}/d) &&\triangleright \textit{Contextual View} \quad (2) \\
s_i^{im} &= \text{Norm}(v_i^T \cdot E_{im}/d) &&\triangleright \textit{Structural View}
\end{aligned}
$$

where $E_{st}$, $E_{dt}$, and $E_{im}$ denote the global embedding vectors of sparse text, dense text, and the global image token, respectively. $\text{Norm}(\cdot)$ represents the min-max normalization to scale values into $[0, 1]$. $d$ is the embedding dimension.

In this formulation, $s_i^{st}$ captures high-frequency alignment (e.g., matching "dog" to the dog patch), while $s_i^{dt}$ captures low-frequency semantic completeness (e.g., matching the grassy field context). $s_i^{dt}$ effectively acts as a semantic regularizer, preventing the model from discarding patches that are visually significant but textually omitted in the sparse caption. The final calibrated score $s_i$ is derived via a weighted consensus mechanism:

$$
\begin{aligned}
s_i^s &= (1 - \beta) \cdot s_i^p + \beta \cdot (s_i^{st} + s_i^{im})/2 \\
s_i^d &= (1 - \beta) \cdot s_i^p + \beta \cdot (s_i^{dt} + s_i^{im})/2
\end{aligned}
\quad (3)
$$

$\beta$, empirically set to 0.6, serves as a gatekeeper, determining how much the semantic consensus can override the initial visual saliency. This weighted combination ensures that the selected patches are robust to the linguistic variations of sparse captions while maintaining visual structural integrity.

**Differentiable Decision and Feature Re-synthesis.** In the second stage, we transform the continuous scores $s \in \mathbb{R}^N$ into discrete selection decisions. A standard Top-K selection operation is non-differentiable, which hinders end-to-end training of the vision encoder. To overcome this, we adopt the Gumbel-Softmax relaxation technique (Maddison et al., 2016) to generate differentiable decision masks.

For the sparse-text guided branch, the decision mask $D_s$ is sampled as:

$$D_s = \text{Softmax}\left(\frac{\log(\pi) + g}{\tau}\right) \quad (4)$$

where $\pi$ represents the class probabilities derived from scores $s^s$, $g$ constitutes independent Gumbel noise samples, and $\tau$ is the temperature parameter controlling the smoothness of the distribution. As $\tau \to 0$, the samples approach a categorical one-hot distribution. We apply the same logic to generate the dense-text guided mask $D_d$. These masks effectively separate the visual input into a set of discriminative patches $V_s$ and contextual patches $V_d$.

Finally, to avoid information fragmentation caused by hard dropping, we propose an Adaptive Aggregation Network that re-synthesizes the selected patches into a compact set of $N_c$ informative tokens. Rather than simple pooling, we employ a learnable linear combination:

$$\hat{v}_j = \sum_{i=1}^{N_s}(W_s)_{ij} \cdot v_i^s + \sum_{i=1}^{N_d}(W_d)_{ij} \cdot v_i^d, \quad (5)$$
$$j \in \{1, \ldots, N_c\}$$

Here, $N_s$ and $N_d$ denote the numbers of selected patches corresponding to different granularities, while the weight matrices $W_s = \text{Softmax}(\text{MLP}(V_s))$ and $W_d = \text{Softmax}(\text{MLP}(V_d))$ serve as dynamic routers. This aggregation step functions as a semantic clustering process, where redundant patch features are merged into representative centroids $\hat{v}_j$. This allows the model to compress the visual sequence length while preserving the semantic richness captured by the dual-granularity scoring.

### 3.3. Salience-Guided Metric Aggregation

Standard mean-pooling alignment is susceptible to noise from irrelevant background patches. The SGMA module introduces a relevance-aware manifold alignment strategy that prioritizes highly-relevant correspondences.

Given the re-synthesized patches $\hat{V}$ and textual tokens $T$, we first compute the fine-grained similarity matrix $A \in$

$\mathbb{R}^{N_c \times M}$. Instead of simple averaging, we decompose the alignment into bi-directional salience accumulation. We identify the maximum response pairs (Hard Alignment) and learn a scalar residual to capture soft alignment cues:

$$S(I,T) = \underbrace{\left(\frac{1}{N_c}\sum_{i=1}^{N_c}\max_j(A)_{ij} + \phi(\text{TOPK}(\max_j(A)_{ij}))\right.}_{\text{Visual-Dominant Alignment}}$$
$$\underbrace{+ \frac{1}{M}\sum_{j=1}^{M}\max_i(A)_{ij} + \phi(\text{TOPK}(\max_i(A)_{ij}))}_{\text{Textual-Dominant Alignment}}$$
$$(6)$$

where $\phi(\cdot)$ denotes the learnable mapping (MLP). This formulation amplifies the contribution of maximally aligned patch-word pairs, making the final similarity score $S(I, T)$ robust against similarity shift bias and patch redundancy.

Following prior work, we adopt a bidirectional triplet loss with hard negative mining(Faghri et al., 2017):

$$\mathcal{L}_{\text{align}} = \sum_{(I,T)}\left(\left[\alpha - S(I,T) + S(I, \hat{T})\right]_+ \right.$$
$$\left. + \left[\alpha - S(I,T) + S(\hat{I}, T)\right]_+\right) \quad (7)$$

where $\alpha$ is the margin, $[x]_+ = \max(x, 0)$, and $(I, T)$ denotes a positive image–text pair within the mini-batch. The hardest negatives are defined as $\hat{T} = \arg\max_{j \neq T} S(I, j)$ and $\hat{I} = \arg\max_{i \neq I} S(i, T)$ for text and image, respectively.

Furthermore, to enhance training stability, we constrain the proportion of selected patches to a target value $\rho$(Rao et al., 2021), and supervise this constraint using mean-squared-error losses computed from the sparse-text and dense-text views, respectively. Finally, we combine the cross-modal alignment loss $\mathcal{L}_{\text{align}}$ Eq.7 with the ratio constraint loss $\mathcal{L}_{\text{ratio}}$:

$$\mathcal{L}_{\text{ratio}} = \left(\rho - \lambda_1 \cdot \frac{1}{N_s}\sum_{i=1}^{N_s}(D_s)_i - \lambda_2 \cdot \frac{1}{N_d}\sum_{i=1}^{N_d}(D_d)_i\right)^2,$$
$$\mathcal{L} = \mathcal{L}_{\text{align}} + \mathcal{L}_{\text{ratio}}$$
$$(8)$$

where $\lambda_1$ and $\lambda_2$ are constant coefficients for sparse text and dense text.

## 4. Experiments

### 4.1. Datasets

Following prior works (Diao et al., 2021; Faghri et al., 2017; Lee et al., 2018), we evaluated this model on the widely-

*Table 1.* Comparisons of image-text retrieval performance on Flickr30K and MS-COCO test-set. We list the details of feature encoding, image resolution, and the number of obtained regions/patches by visual encoder (e.g. "ViT-Base-224" represents the base-version of Vision Transformer with 224×224 image resolution input, regarding 16×16 pixels as one patch, and getting 14×14 visual patches for one image). FG indicates whether it is the fine-grained cross-modal alignment. The best results are marked **bold**, and the second best results are marked underlined.

| Method | FG | Flickr30K 1K I2T R@1 | R@5 | R@10 | T2I R@1 | R@5 | R@10 | rSum | MS-COCO 1K I2T R@1 | R@5 | R@10 | T2I R@1 | R@5 | R@10 | rSum | MS-COCO 5K I2T R@1 | R@5 | R@10 | T2I R@1 | R@5 | R@10 | rSum |
|---|---|---|---|---|---|---|---|---|---|---|---|---|---|---|---|---|---|---|---|---|---|---|
| *ViT-Base-224 + BERT-base, 14×14 patches* | | | | | | | | | | | | | | | | | | | | | | |
| VSE++ (Faghri et al., 2017) | ✗ | 71.8 | 92.8 | 96.5 | 59.4 | 84.7 | 90.9 | 496.1 | 75.0 | 94.6 | 98.0 | 62.7 | 89.4 | 94.9 | 514.6 | 52.4 | 80.3 | 88.8 | 40.6 | 70.4 | 81.1 | 413.4 |
| SCAN (Lee et al., 2018) | ✓ | 69.5 | 90.9 | 95.6 | 56.4 | 83.1 | 90.0 | 485.6 | 76.0 | 95.4 | 98.1 | 64.5 | 90.8 | 95.8 | 520.6 | 53.9 | 81.8 | 90.0 | 42.9 | 72.3 | 82.5 | 423.5 |
| SGR (Diao et al., 2021) | ✓ | 69.7 | 90.8 | 95.2 | 59.1 | 84.1 | 89.9 | 488.7 | 77.2 | 95.0 | 98.0 | 65.1 | 90.7 | 95.8 | 521.8 | 54.9 | 82.8 | 90.5 | 42.8 | 72.2 | 82.5 | 425.8 |
| CHAN (Pan et al., 2023) | ✓ | 69.2 | 91.8 | 95.0 | 58.4 | 84.9 | 90.6 | 489.9 | 77.1 | 95.1 | 98.1 | 65.0 | 91.0 | 96.0 | 522.2 | 56.3 | 83.2 | 90.1 | 43.0 | 72.6 | 82.8 | 428.0 |
| LAPS (Fu et al., 2024) | ✓ | 74.0 | 93.4 | 97.4 | 62.5 | 87.3 | 92.7 | 507.3 | 78.7 | 95.5 | 98.3 | 66.2 | 91.3 | 96.2 | 526.3 | 57.5 | 84.0 | 90.8 | 44.5 | 74.0 | 83.6 | 434.4 |
| AVSE (Liu et al., 2025d) | ✗ | 76.0 | 94.6 | 97.5 | 62.7 | 88.4 | 93.1 | 512.3 | 79.8 | 95.6 | 98.3 | 67.0 | 91.5 | 96.3 | 528.5 | 58.8 | 84.3 | 91.0 | 45.1 | 74.3 | 83.9 | 437.4 |
| D2S-VSE (Liu et al., 2025c) | ✗ | 82.8 | **96.1** | **98.3** | 68.5 | 91.3 | 94.9 | 531.9 | 80.1 | **97.0** | **99.2** | 68.1 | 92.5 | 96.7 | 533.7 | 60.1 | **85.5** | **92.5** | 46.3 | 75.9 | 85.2 | 445.6 |
| **SEPS** | ✓ | **86.1** | 93.7 | 96.9 | **86.9** | **98.1** | **99.2** | **560.9** | **89.0** | 94.8 | 98.0 | **88.5** | **99.3** | **99.8** | **569.5** | **73.9** | 85.2 | 92.1 | **73.5** | **94.5** | **97.8** | **516.9** |
| *ViT-Base-384 + BERT-base, 24×24 patches* | | | | | | | | | | | | | | | | | | | | | | |
| VSE++ (Faghri et al., 2017) | ✗ | 77.1 | 95.7 | 97.5 | 65.8 | 90.2 | 94.3 | 520.5 | 77.0 | 95.7 | 98.4 | 64.6 | 91.1 | 96.2 | 523.0 | 54.9 | 82.8 | 90.4 | 42.4 | 72.4 | 82.8 | 425.8 |
| SCAN (Lee et al., 2018) | ✓ | 75.4 | 94.4 | 96.9 | 63.6 | 88.6 | 93.5 | 512.5 | 76.1 | 95.5 | 98.5 | 65.1 | 91.6 | 96.3 | 523.1 | 53.3 | 81.8 | 90.0 | 42.6 | 72.6 | 82.9 | 423.1 |
| SGR (Diao et al., 2021) | ✓ | 76.9 | 94.9 | 98.1 | 64.2 | 88.4 | 93.3 | 515.8 | 75.8 | 95.7 | 98.6 | 65.6 | 92.0 | 96.5 | 524.2 | 53.3 | 81.0 | 89.6 | 42.9 | 73.1 | 83.7 | 423.6 |
| CHAN (Pan et al., 2023) | ✓ | 75.4 | 94.5 | 97.6 | 63.2 | 88.6 | 93.1 | 512.4 | 78.1 | 95.8 | 98.6 | 66.1 | 92.1 | 96.6 | 527.3 | 55.6 | 83.8 | 91.2 | 43.4 | 73.6 | 83.5 | 431.1 |
| LAPS (Fu et al., 2024) | ✓ | 79.0 | 96.0 | 98.1 | 67.3 | 90.5 | 94.5 | 525.4 | 78.6 | 96.3 | 98.9 | 68.0 | 92.4 | 96.8 | 531.0 | 57.4 | 84.9 | 92.5 | 46.4 | 75.8 | 85.2 | 442.2 |
| AVSE (Liu et al., 2025d) | ✗ | 80.3 | 96.4 | 98.7 | 67.9 | 91.2 | 94.7 | 529.2 | 81.1 | 97.1 | 99.0 | 68.3 | 92.7 | 97.0 | 535.2 | 61.2 | 86.8 | 93.2 | 46.2 | 75.9 | 85.0 | 448.3 |
| D2S-VSE (Liu et al., 2025c) | ✗ | 84.1 | **97.5** | **99.1** | 70.3 | 91.6 | 95.3 | 537.9 | 80.8 | **97.2** | **99.1** | 69.0 | 92.9 | 96.8 | 535.8 | 60.6 | 86.5 | 93.2 | 46.8 | 76.4 | 85.7 | 449.1 |
| **SEPS** | ✓ | **90.7** | 94.4 | 98.4 | **89.3** | **99.3** | **99.5** | **571.5** | **90.9** | 96.1 | 98.8 | **91.0** | **99.5** | **99.8** | **576.1** | **77.8** | **88.7** | **94.8** | **78.5** | **96.3** | **98.7** | **534.6** |
| *Swin-Base-224 + BERT-base, 7×7 patches* | | | | | | | | | | | | | | | | | | | | | | |
| VSE++ (Faghri et al., 2017) | ✗ | 82.5 | 96.5 | 98.9 | 70.0 | 91.4 | 95.1 | 534.4 | 83.3 | 97.5 | 99.3 | 71.0 | 93.0 | 96.7 | 540.9 | 64.0 | 88.2 | 94.2 | 49.9 | 78.0 | 86.6 | 460.9 |
| SCAN (Lee et al., 2018) | ✓ | 79.0 | 95.9 | 98.2 | 67.7 | 90.6 | 94.9 | 526.3 | 80.9 | 97.0 | 99.1 | 69.7 | 93.1 | 97.1 | 536.9 | 60.7 | 86.6 | 93.2 | 48.1 | 77.1 | 86.1 | 451.8 |
| SGR (Diao et al., 2021) | ✓ | 80.4 | 97.0 | 98.7 | 66.9 | 90.2 | 94.5 | 527.6 | 81.2 | 97.1 | 99.1 | 69.9 | 93.2 | 97.2 | 537.7 | 61.0 | 86.7 | 93.2 | 48.6 | 77.2 | 86.3 | 453.1 |
| CHAN (Pan et al., 2023) | ✓ | 81.4 | 97.0 | 98.6 | 68.5 | 90.6 | 94.5 | 530.6 | 81.6 | 97.2 | 99.3 | 70.6 | 93.7 | 97.6 | 539.8 | 64.1 | 87.9 | 93.5 | 49.1 | 77.3 | 86.1 | 458.0 |
| LAPS (Fu et al., 2024) | ✓ | 82.4 | 97.4 | 99.5 | 70.0 | 91.7 | 95.4 | 536.3 | 84.0 | 97.6 | 99.3 | 72.1 | 93.7 | 97.3 | 544.1 | 64.5 | 89.2 | 94.4 | 51.6 | 78.9 | 87.2 | 465.8 |
| AVSE (Liu et al., 2025d) | ✗ | 83.9 | 97.4 | 99.4 | 70.0 | 92.4 | 95.6 | 538.7 | 84.9 | **98.0** | 99.3 | 72.1 | 94.0 | 97.4 | 545.7 | 66.2 | 89.8 | 94.7 | 51.7 | 79.2 | 87.3 | 468.9 |
| D2S-VSE (Liu et al., 2025c) | ✗ | 87.2 | 98.4 | **99.9** | 73.0 | 93.5 | 96.7 | 548.7 | 82.4 | 97.6 | **99.3** | 70.3 | 93.7 | 97.4 | 540.7 | 63.9 | 87.7 | 94.0 | 49.3 | 78.3 | 87.2 | 460.4 |
| **SEPS** | ✓ | **89.8** | 96.9 | 98.7 | **88.0** | **98.9** | **99.6** | **572.0** | **87.2** | 94.9 | 98.3 | **84.7** | **99.0** | **99.8** | **563.9** | **71.9** | 86.0 | 92.4 | **66.8** | **92.2** | **96.8** | **506.1** |
| *Swin-Base-384 + BERT-base, 12×12 patches* | | | | | | | | | | | | | | | | | | | | | | |
| VSE++ (Faghri et al., 2017) | ✗ | 83.8 | 97.5 | 99.2 | 71.1 | 93.2 | 96.2 | 540.6 | 82.9 | 97.7 | 99.4 | 71.3 | 93.5 | 97.3 | 542.1 | 63.0 | 88.5 | 94.3 | 50.1 | 78.9 | 87.4 | 462.2 |
| SCAN (Lee et al., 2018) | ✓ | 81.9 | 96.9 | 98.9 | 70.0 | 92.7 | 95.8 | 536.1 | 81.6 | 96.8 | 99.1 | 69.1 | 92.7 | 96.7 | 536.1 | 61.1 | 87.3 | 93.3 | 47.8 | 76.9 | 85.9 | 452.4 |
| SGR (Diao et al., 2021) | ✓ | 80.7 | 96.8 | 99.0 | 69.9 | 91.7 | 95.3 | 533.4 | 81.9 | 96.7 | 99.1 | 69.3 | 92.8 | 96.7 | 536.6 | 62.8 | 87.0 | 92.9 | 48.1 | 77.0 | 86.0 | 453.8 |
| CHAN (Pan et al., 2023) | ✓ | 81.2 | 96.7 | 98.8 | 70.3 | 92.2 | 95.9 | 535.0 | 83.1 | 97.3 | 99.2 | 70.4 | 93.1 | 97.1 | 540.2 | 63.4 | 88.4 | 94.1 | 49.2 | 77.9 | 86.6 | 459.5 |
| LAPS (Fu et al., 2024) | ✓ | 85.1 | 97.7 | 99.2 | 74.0 | 93.0 | 96.3 | 545.3 | 84.1 | 97.4 | 99.2 | 72.1 | 93.9 | 97.4 | 544.1 | 67.1 | 88.6 | 94.3 | 53.0 | 79.5 | 87.6 | 470.1 |
| AVSE (Liu et al., 2025d) | ✗ | 87.1 | 98.3 | 99.2 | 73.6 | 93.5 | 96.5 | 548.2 | 85.1 | **98.2** | **99.5** | 71.6 | 94.0 | 97.5 | 545.9 | 68.6 | **90.2** | **95.6** | 52.2 | 79.6 | 87.8 | **474.0** |
| D2S-VSE (Liu et al., 2025c) | ✗ | 87.8 | **99.0** | **99.7** | 75.7 | 94.1 | 96.9 | 553.2 | 83.8 | 97.9 | 99.4 | 71.9 | 94.2 | 97.9 | 544.7 | 65.2 | 89.2 | 94.6 | 51.3 | 79.4 | 87.9 | 467.7 |
| **SEPS** | ✓ | **93.6** | 98.3 | 99.2 | **91.6** | **99.4** | **99.8** | **581.9** | **89.5** | 96.5 | 99.0 | **87.1** | **99.2** | **99.9** | **571.2** | **74.7** | 88.4 | 94.3 | **70.3** | **93.8** | **97.6** | **519.1** |

used Flickr30K (Young et al., 2014) and MS-COCO (Lin et al., 2014) benchmarks. Each image in these datasets is paired with five textual captions. For Flickr30K, we adopted the standard split of 29,000 training, 1,000 validation, and 1,014 test images. For MS-COCO, we used the common split of 113,287 for training, 5,000 for validation, and 5,000 for testing. We reported results on both the 1K test set (averaged over 5 folds) and the full 5K test set.

### 4.2. Metrics

We adopted Recall@K (R@K, $K \in \{1, 5, 10\}$) and rSum as evaluation metrics. R@K measures the percentage of ground truth in the retrieved top-K lists, while rSum aggregates multiple R@K in both directions (image-to-text and text-to-image) to summarize overall retrieval quality. Our code is based on the public code of LAPS (Fu et al., 2024).

The dense text generation was followed by D2S-VSE (Liu et al., 2025c).

### 4.3. Comparison with State-of-the-art Methods

Following the standard protocols of two benchmarks (Faghri et al., 2017; Zhang et al., 2022), we systematically compared the retrieval performance of SEPS with recent state-of-the-art methods on Flickr30K and MS-COCO. Table 1 detailed the feature encoders, input resolutions, and whether fine-grained alignment (FG) was adopted for each method. The performance of competing methods was reported directly from their original publications, supplementing with ensemble versions where necessary for comparison. Firstly, we introduce three SOTA cross-modal alignment methods:

• **LAPS (Fu et al., 2024):** A fine-grained approach that

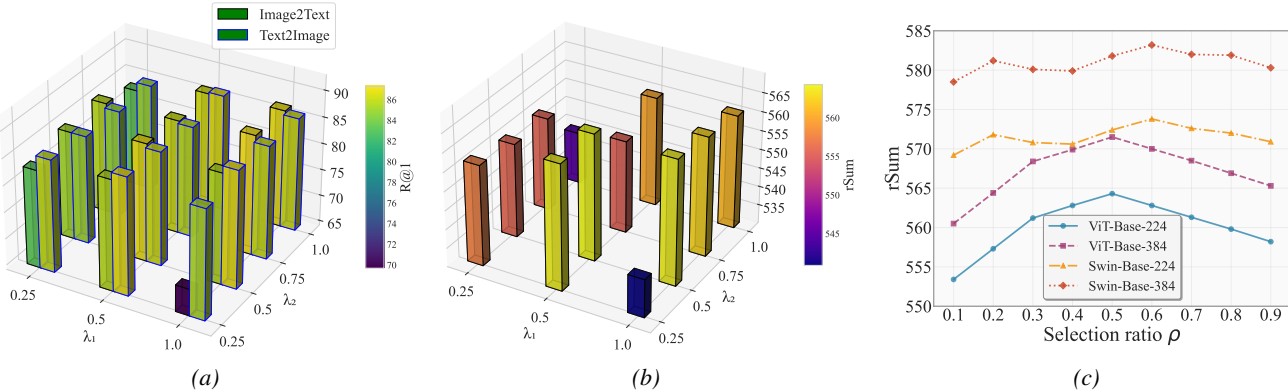

*Figure 3.* The retrieval performance of different selection ratios $\rho$, constant coefficients $\lambda_1$ and $\lambda_2$ with various visual encoders on Flickr30K.

*Table 2.* The comparisons of image-text retrieval for Vision-Language Pre-training (VLP) Models. *FG* indicates whether the method fine-grained alignment. * means the zero shot learning.

| Method | FG | Flickr30K 1K | | | | MS-COCO 5K | | | |
|---|---|---|---|---|---|---|---|---|---|
| | | Image-to-Text | | Text-to-Image | | Image-to-Text | | Text-to-Image | |
| | | R@1 | R@5 | R@1 | R@5 | R@1 | R@5 | R@1 | R@5 |
| *CLIP-ViT-Base-224 + CLIP-BERT-Base, $14 \times 14$ patches* | | | | | | | | | |
| CLIP* (Radford et al., 2021) | ✗ | 81.4 | 96.2 | 61.1 | 85.4 | 52.3 | 76.2 | 33.3 | 58.2 |
| VSE++ (Faghri et al., 2017) | ✗ | 92.2 | 99.1 | 80.5 | 95.6 | 68.0 | 88.2 | 53.6 | 79.7 |
| SCAN (Lee et al., 2018) | ✓ | 88.2 | 98.1 | 75.3 | 93.1 | 65.4 | 88.0 | 50.7 | 77.6 |
| LAPS (Fu et al., 2024) | ✓ | 92.9 | 99.3 | 80.6 | 95.5 | 69.8 | 90.4 | 54.3 | 80.0 |
| **SEPS** | ✓ | **94.7** | 97.6 | **93.1** | **97.7** | **84.1** | **91.2** | **78.4** | **95.5** |
| *CLIP-ViT-Large-224 + CLIP-BERT-Large, $16 \times 16$ patches* | | | | | | | | | |
| CLIP* (Radford et al., 2021) | ✗ | 85.0 | 97.7 | 61.3 | 87.0 | 55.9 | 79.1 | 35.9 | 60.9 |
| VSE++ (Faghri et al., 2017) | ✗ | 94.0 | 99.5 | 83.4 | 96.4 | 68.5 | 89.4 | 56.7 | 81.9 |
| SCAN (Lee et al., 2018) | ✓ | 90.0 | 98.5 | 82.0 | 95.9 | 68.0 | 90.4 | 53.2 | 80.7 |
| LAPS (Fu et al., 2024) | ✓ | 94.6 | 99.9 | 84.9 | 97.3 | 72.9 | 91.7 | 57.1 | 81.3 |
| **SEPS** | ✓ | **95.8** | 98.4 | **95.1** | **98.1** | **86.5** | 91.7 | **79.3** | **95.8** |

prunes redundant patches under language guidance, followed by semantic and spatial calibration to enable sparse, bidirectional patch–word alignment.

• **AVSE** (Liu et al., 2025d): A coarse-grained approach that constructs multi-view global image embeddings via radial-biased sampling and performs Asymmetric Embedding Optimal Matching (AEOM) for global alignment.

• **D2S-VSE** (Liu et al., 2025c): A coarse-grained approach that leverages dense-to-sparse distillation with dense captions generated by a multimodal large language model (MLLM) to align cross-modal information capacity, and conducts retrieval via global embedding similarity.

**Image-text Retrieval Performance.** The quantitative results in Table 1 confirm that SEPS establishes a new state-of-the-art benchmark across all evaluated protocols. The substantial improvements in text-to-image retrieval stem from our resolution of *Semantic Sparsity Bias*—sparse textual queries often contain ambiguous references that fail to specify which visual regions should be matched, making it difficult to distinguish among visually similar candidates. By introducing the the semantic anchor, SEPS disambiguates these references through dense semantic context, enabling precise patch-word correspondences. Consistent improve-

ments across all backbones further validate the robustness of our semantic consensus mechanism.

*Table 3.* The zero-shot evaluation on image-text retrieval task. All models are trained by CLIP backbones of ViT-B/16 in Flickr dataset([#] is untrained), and evaluated in MS-COCO dataset.

| Method | FG | MS-COCO 1K | | | | MS-COCO 5K | | | |
|---|---|---|---|---|---|---|---|---|---|
| | | Image-to-Text | | Text-to-Image | | Image-to-Text | | Text-to-Image | |
| | | R@1 | R@5 | R@1 | R@5 | R@1 | R@5 | R@1 | R@5 |
| CLIP[#] (Radford et al., 2021) | ✗ | - | - | - | - | 52.3 | 76.2 | 33.3 | 58.2 |
| VSE++ (Faghri et al., 2017) | ✗ | 28.1 | 56.4 | 20.5 | 46.8 | 13.2 | 30.5 | 9.1 | 24.6 |
| D2S-VSE (Liu et al., 2025c) | ✗ | 32.3 | 62.3 | 24.8 | 53.9 | 15.9 | 34.7 | 11.3 | 28.3 |
| SCAN (Lee et al., 2018) | ✓ | 30.5 | 59.8 | 23.2 | 51.5 | 14.8 | 32.9 | 10.5 | 26.8 |
| LAPS (Fu et al., 2024) | ✓ | 47.4 | 74.9 | 35.8 | 66.1 | 27.1 | 50.5 | 19.0 | 50.5 |
| **SEPS** | ✓ | **65.8** | **77.4** | **62.8** | **87.5** | 45.8 | 60.2 | **44.3** | **69.5** |

*Table 4.* The zero-shot evaluation on visual grounding task. All models are trained by CLIP backbones of ViT-B/16 in Flickr dataset ([#] is untrained). Following ReCLIP (Subramanian et al., 2022), we apply the Grad-GAM (Selvaraju et al., 2020) to select the bounding box from proposals.

| Models | FG | RefCOCO | | | RefCOCO+ | | | RefCOCOg | |
|---|---|---|---|---|---|---|---|---|---|
| | | Val | TestA | TestB | Val | TestA | TestB | Val | Test |
| CLIP[#] (Radford et al., 2021) | ✗ | 39.3 | 45.3 | 34.2 | 41.2 | 47.0 | 36.8 | 45.0 | 45.9 |
| VSE++ (Faghri et al., 2017) | ✗ | 40.7 | 46.3 | 33.6 | 43.2 | 49.0 | 35.6 | 44.2 | 43.9 |
| SCAN (Lee et al., 2018) | ✓ | 41.8 | 47.3 | 44.4 | 43.2 | 49.3 | 36.8 | 45.2 | 46.0 |
| LAPS (Fu et al., 2024) | ✓ | 44.2 | 49.9 | 38.4 | 46.7 | 52.3 | 41.6 | 51.3 | 51.2 |
| **SEPS** | ✓ | **48.7** | **52.3** | **43.4** | **51.2** | **54.8** | **46.6** | **55.3** | **55.2** |

**Compatibility with VLP Models.** To demonstrate the universality of SEPS beyond task-specific fine-tuning, we extend the framework to the widely adopted CLIP architecture (Radford et al., 2021). As detailed in Table 2, SEPS successfully enhances the representational quality of VLP models. This confirms that the Visual-Linguistic Information Asymmetry is a fundamental bottleneck inherent in multimodal learning, not an artifact of specific architectures.

**Zero-shot Transfer Capability.** Table 3 presents zero-shot evaluations where models trained on Flickr30K are directly deployed on MS-COCO without fine-tuning. SEPS substantially outperforms all fine-grained baselines, exhibiting superior generalization capabilities. We attribute this to the semantic consensus voting mechanism, which forces the model to learn intrinsic visual-semantic correspondences.

*Table 5.* Ablation study of different modules for SEPS framework on Flickr30K.

| Modules | Different Settings | Image-to-Text R@1 | Image-to-Text R@5 | Text-to-Image R@1 | Text-to-Image R@5 |
|---|---|---|---|---|---|
| DGSC | Only sparse text | 78.6 | 90.1 | 67.2 | 88.5 |
| | Only dense text | 80.3 | 92.4 | 80.5 | 91.8 |
| | concat(sparse, dense) | 80.9 | 92.5 | 80.8 | 92.2 |
| | sparse_score + dense_score | 82.5 | 93.0 | 81.1 | 92.4 |
| | Without aggregation | 84.2 | 93.3 | 83.7 | 94.3 |
| SGMA | Only relevance-aware selection | 85.3 | 93.5 | 84.1 | 94.9 |
| | Only mean value | 83.5 | 92.6 | 82.4 | 93.5 |
| | Complete **SEPS** | **86.1** | **93.7** | **86.9** | **98.1** |

*Table 6.* Comparison of computational costs across different methods. Inference cost is measured per image-text pair on a single NVIDIA A100 GPU.

| Method | Offline Preprocess | Online Inference |
|---|---|---|
| LAPS (Fu et al., 2024) | 0 | 92ms |
| D2S-VSE (Liu et al., 2025c) | LLaVA 8s/image | 78ms |
| SEPS | LLaVA 8s/image | 107ms |

**Extension to Visual Grounding.** As demonstrated in Table 4, SEPS consistently surpasses baselines, with particularly notable gains on the challenging RefCOCO+ dataset, which forbids location-based expressions. This evidences that the rich semantic representations acquired through our calibration mechanism effectively transfer to localization tasks without requiring explicit spatial supervision.

### 4.4. Ablation Study

We conducted comprehensive ablation studies on Flickr30K to systematically evaluate the contribution of each component.

**Impact of Dual-Granularity Integration.** As detailed in Table 5, while utilizing dense captions alone surpasses the sparse baseline, naive fusion strategies yield suboptimal improvements. We attribute this to *Semantic Drift*, where uncalibrated dense signals introduce noise that conflicts with specific queries. In contrast, the full DGSC module achieves substantial gains, empirically proving that our mechanism does not merely accumulate features but performs a strategic semantic consensus.

**Computational Efficiency Analysis.** Table 6 reveals that SEPS incurs only a marginal overhead in inference latency while sharing identical preprocessing costs with SOTA methods like D2S-VSE. This indicates that the framework maintains acceptable computational efficiency for practical deployment.

**Hyperparameter Robustness.** Figure 3 illustrates that SEPS maintains consistent performance stability. This insensitivity suggests that the improvements stem from the robust theoretical foundation of the semantic consensus voting mechanism.

*Table 7.* Ablation study on dense text sources (Flickr30K, ViT-Base-224).Different dataset refers to MS-COCO dataset, and different image means random choice for each image-text pair.

| Different Dense-Text Sources | I2T | T2I |
|---|---|---|
| Different dataset, Different image | 77.2 | 71.7 |
| Same dataset, Different image | 80.8 | 79.3 |
| Same dataset, Same image | **86.1** | **86.9** |

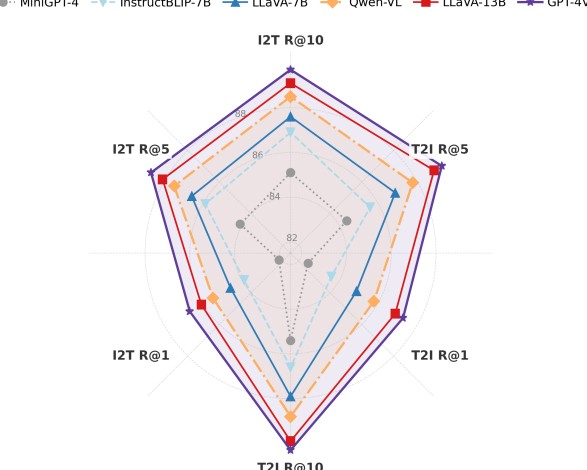

*Figure 4.* Ablation study on different MLLMs for dense text generation (Flickr30K, ViT-Base-224).

**Dense Text Source.** Table 7 demonstrated that the performance gains stem from the rich semantic information inherent in dense text. This confirmed that the model effectively leverages the detailed linguistic anchor to enhance performance in the image-text alignment.

**Influence of MLLM Capacity.** Figure 4 investigates the impact of different MLLM backbones on the Holistic Anchor generation. Notably, performance gains saturate around LLaVA-13B. It confirms that the efficacy of SEPS derives principally from the structural innovations of the DGSC and SGMA modules.

## 5. Conclusion

This work introduces SEPS to resolve the **Semantic Sparsity Bias** in cross-modal alignment. We propose a structure-centric approach where the DGSC mechanism synthesizes a Holistic Visual-Linguistic Anchor via MLLMs, establishing a semantic consensus that filters patch redundancy. Furthermore, the SGMA mitigates similarity dilution in traditional metrics. Extensive experiments confirm that SEPS sets new state-of-the-art benchmarks, demonstrating robust zero-shot generalization and offering a scalable paradigm for multimodal representation learning. Future work could extend this dual-text paradigm to broader multimodal tasks and leverage advances in MLLMs for enhanced cross-modal alignment.

## Acknowledgements

This work was supported in part by Sichuan Province Science and Technology Support Program under Grant No. 2025ZDZX0016.

## Impact Statement

This work may improve fine-grained image–text retrieval and multimodal grounding by selecting more relevant visual evidence. It also inherits risks from MLLM-generated dense text, including hallucinated attributes, demographic bias, privacy-sensitive inferences, and potential misuse of retrieval systems for surveillance or profiling. We therefore recommend auditing generated descriptions and retrieval behavior across deployment domains and using SEPS only with appropriate privacy, consent, and fairness safeguards.

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
