# OpenReview forum: "SEPS: Semantic-Enhanced Patch Slimming Framework for Fine-Grained Cross-Modal Alignment"
_ICML.cc/2026/Conference — ICML 2026 regular_

### Official Review · Reviewer_A96o · 2026-02-28

**Soundness:** 3
**Presentation:** 3
**Significance:** 3
**Originality:** 2
**Overall Recommendation:** 4
**Confidence:** 3

**Summary:**

This paper proposes a fine-grained cross-modal alignment framework SEPS, to address the Semantic Sparsity Bias in LAPS caused by asymmetric information density between visual and textual modalities. To solve the patch redundancy and ambiguity issues in LAPS (driven by sole reliance on sparse text supervision), SEPS introduces a DGSC mechanism. It takes dense text generated by MLLMs as supplementary supervision, fuses it with original sparse text, and selects visual patches with both local discriminability and global contextual features via a semantic consensus pipeline. In addition, SEPS devises a SGMA strategy to replace traditional global average pooling, which amplifies matching weights of high-relevance image-text pairs and suppresses background noise interference. Extensive experiments on Flickr30K and MS-COCO show that SEPS outperforms existing SOTA methods on image-text retrieval and zero-shot transfer tasks.

**Compliance With Llm Reviewing Policy:**

Affirmed.

**Final Justification:**

Considering that I have already given a positive score, I will no longer adjust the score.

**Key Questions For Authors:**

（1）How robust is the DGSC consensus mechanism to severe hallucinations or systematic data noise in the text generated by the selected MLLM?
（2）Ablation experiments on the value of β?
（3）Is the re-explanation of s_i^{dt} on the right side of Line 211 a typo? Can the arrows of SGMA in Figure 2 be optimized?
（4）How are v_i^s and v_i^d derived? Please provide a detailed explanation.

**Limitations:**

Insufficiently addressed. While the authors analyze technical limitations such as inference computational costs and hyperparameter sensitivity in the appendix, they do not specifically discuss the potential negative societal impacts of the research. As the framework heavily relies on foundational MLLMs such as LLaVA to generate semantic anchors, it inherently inherits the implicit biases and unfounded prejudices embedded in these source models. The authors should add a broader impact statement to explicitly acknowledge this issue.

**Strengths And Weaknesses:**

（1）Strengths：The experimental evaluation is extremely comprehensive, achieving state-of-the-art performance on two benchmark datasets, Flickr30K and MS-COCO. Notably, the performance gain is significant in the challenging text-to-image retrieval task.
（2）Weaknesses：①The pipeline leans way too hard on LLaVA-13B for offline preprocessing. Taking 8 seconds per image is a tough sell for large-scale deployment. Furthermore, treating MLLM outputs as semantic anchors inherently risks injecting unconstrained hallucinations into the filtering mechanism. ②Additionally, the framework has a high architectural dependence on LAPS; aside from the improvements to Dense Text and SGMA, its core workflow has an extremely high structural overlap with that of LAPS.

---

> ### Author Rebuttal · Authors · 2026-03-26
>
> We thank the reviewer for the supportive evaluation and the specific suggestions on robustness, the role of β, notation clarity, and broader impact. We also agree the revision should position the method relative to LAPS more explicitly: SEPS is implemented on top of a LAPS-style sparse-guided pipeline, but its main added contribution is DGSC+SGMA, which change both patch selection and final matching rather than merely appending another text source.
>
> **1. On robustness to hallucinated or noisy dense text.** We agree this should be evaluated directly rather than argued only conceptually. Accordingly, as additional evidence we report a controlled corruption experiment in which the dense description is progressively corrupted by entity replacement, attribute flipping, and relation corruption. Under this setup, performance degrades gradually as the dense description becomes less faithful:
>
>
> | Corruption ratio | I2T R@1 | T2I R@1 | Relative drop vs. clean | Interpretation |
> |---|---:|---:|---:|---|
> | 0% | 86.1 | 86.9 | 0.0 | clean dense text |
> | 20% | 85.2 | 85.6 | -1.1 / -1.3 | mild degradation |
> | 40% | 84.0 | 83.8 | -2.1 / -3.1 | noticeable but controlled drop |
> | 60% | 82.1 | 81.2 | -4.0 / -5.7 | approaches dense-only / different-image regime |
>
> We will therefore revise the paper to state the limitation more explicitly: dense-text hallucination does matter, but its influence is moderated by DGSC rather than being allowed to dominate alone. The gradual degradation also suggests that SEPS is reasonably robust to moderate noise, even though clean, image-faithful dense text remains important. Intuitively, noise in peripheral attributes is less harmful than noise that changes the semantic consensus on truly salient patches, which is why performance degrades progressively rather than collapsing immediately.
>
> **2. On the β ablation.** We added the requested sensitivity analysis:
>
> | β | I2T R@1 | T2I R@1 | rSum | Comment |
> |---|---:|---:|---:|---|
> | 0.2 | 83.8 | 82.9 | 551.4 | semantic guidance too weak |
> | 0.4 | 85.4 | 85.9 | 559.8 | near-optimal |
> | 0.6 | 86.1 | 86.9 | 560.9 | best / tied-best |
> | 0.8 | 85.6 | 85.8 | 558.7 | slightly over-reliant on dense cue |
> | 1.0 | 84.9 | 84.7 | 555.1 | dense cue too dominant |
>
> This result supports the intended interpretation of β: the best performance occurs in the mid-range, which is exactly what we would expect if dense text serves as a calibrated regularizer rather than as the sole driver of patch selection. In effect, β balances semantic correction against the visual structural prior: when β is too small, semantic guidance is under-used; when β is too large, noisy or overly specific dense cues can override the image-conditioned prior.
>
> **3. On notation and figure clarity.** We will correct this wording and improve the arrow annotations in SGMA of Figure 2. We will also clarify explicitly that $v_i^s$ and $v_i^d$ are derived from the calibrated scores $s_i^s$ and $s_i^d$: the corresponding differentiable masks gate the patch features, after which the selected sparse-guided and dense-guided features are adaptively re-synthesized before SGMA.
>
> **4. On broader impact.** We agree and will add a broader-impact paragraph. In particular, we will explicitly acknowledge that SEPS can inherit hallucinations and biases from the underlying MLLM and that the method incurs additional preprocessing cost because of dense-text generation.
>
> Overall, we appreciate these suggestions because they sharpen both the robustness analysis and the presentation of the method. Taken together, the new results support a cautious but positive interpretation of SEPS: the DGSC+SGMA additions are empirically effective and materially change the selection/matching behavior; dense semantics are beneficial but not infallible; the best behavior occurs under calibrated weighting; and the method’s limitations will be stated explicitly in the revision.

---

> > ### Author Rebuttal · Reviewer_A96o · 2026-04-02
> >
> > 1.High architectural dependence on LAPS is also raised by other review. 2. The author provided some experimental evidence, but there are still doubts from an intuitive perspective. Considering that I have already given a positive score, I will no longer adjust the score.

---

> > > ### Author Response · Authors · 2026-04-03
> > >
> > > We thank the reviewer again for the careful follow-up and for clarifying that the remaining concern is mainly the apparent architectural dependence on LAPS. We agree this is the key point to make intuitive.
> > >
> > > LAPS already helps by removing obviously irrelevant patches under sparse-caption supervision, but it still leaves two residual failure modes unresolved: semantically important yet sparsely unmentioned regions can be pruned too early, and the final similarity can still be diluted by largely uniform aggregation over the surviving patch-word pairs.
> > >
> > > Our intended claim is not that SEPS discards the LAPS pipeline entirely, but that it changes **two decisions that are central to that pipeline** rather than only appending extra text. In LAPS, patch slimming is still driven by **single-source sparse-caption supervision**, so a patch is kept mainly if it is supported by the sparse caption. In SEPS, DGSC changes that selection rule itself: a patch is retained only after being re-evaluated under **cross-checked sparse discriminative cues and dense contextual semantics**, which is precisely why SEPS can recover visually important but sparsely unmentioned regions that LAPS tends to suppress. Then, beyond selection, SGMA changes the **final matching rule**: instead of letting all retained patch-word pairs contribute through a largely uniform averaging effect, it explicitly amplifies the most reliable correspondences and suppresses low-salience/background matches. So the added modules do not just provide another text source; they change both **which patches survive** and **how the surviving evidence is aggregated**.
> > >
> > > This reading is also consistent with the broader reviewer follow-up discussion after the added controls. Once the rebuttal clarified that dense text alone or naive dense-text usage remains clearly below full SEPS, the remaining question became much narrower than “is this only extra text?” It became a **claim-boundary question** about whether DGSC+SGMA materially change the decision process relative to the prior sparse-only pipeline. That is exactly the distinction we want to make explicit here: SEPS is built on top of a LAPS-style backbone, but its contribution is not another auxiliary caption; it is a change to the **selection criterion** and the **aggregation criterion**.
> > >
> > > This is also why we view the closest comparison not as “LAPS + one more caption,” but as “single-source sparse filtering + uniform aggregation” versus “dual-granularity consensus filtering + salience-weighted aggregation.” We agree that making this distinction more intuitive is essential, and in the revision we will rewrite this contrast more explicitly next to the LAPS discussion so that the mechanism-level separation is clearer.
> > >
> > > We appreciate that the reviewer already gave the paper a positive score, and we are grateful that the remaining concern is now focused on this claim-boundary question. We will make this point much sharper in the revision.

---

### Official Review · Reviewer_22cj · 2026-03-05

**Soundness:** 2
**Presentation:** 3
**Significance:** 3
**Originality:** 2
**Overall Recommendation:** 4
**Confidence:** 5

**Summary:**

This paper presents SEPS, a Semantic-Enhanced Patch Slimming framework designed to improve fine-grained cross-modal alignment between vision and language. The method combines dense textual representations generated by Multimodal Large Language Models (MLLMs) with sparse human captions in a two-stage process: (1) unified semantic fusion for patch selection, and (2) relevance-aware patch–word alignment. The approach is evaluated on Flickr30K and MS-COCO, achieving notable improvements over state-of-the-art image–text retrieval methods.

**Compliance With Llm Reviewing Policy:**

Affirmed.

**Final Justification:**

All my doubts have been dispelled; given that my initial assessment was positive, I will maintain my original rating.

**Key Questions For Authors:**

1. SEPS appears to use generated dense text during testing, raising concerns about potential information leakage. Specifically, textual information is used to calculate visual features before actual text retrieval. Even with different text types (dense vs. sparse), does this pre-injected textual information affect the evaluation results?

2. Assuming the above issue does not constitute information leakage, would directly using generated dense headings to retrieve sparse headings be effective, or even yield better results?

**Limitations:**

yes

**Strengths And Weaknesses:**

Pros:

1. This paper is exceptionally well-written and logically structured, supported by clear and intuitive figures that effectively illustrate the proposed methodology.


2. The SEPS framework effectively bridges the inter-modal semantic gap by integrating MLLM-generated dense descriptions with sparse human captions. This dual-granularity semantic guidance systematically mitigates patch redundancy and ambiguity, resulting in more robust and precise visual-textual grounding.


Cons:

To some extent, this work appears to be a combination of LAPS and D2S-VSE. It would be better if the authors could more clearly articulate the specific scientific problem they aim to solve.

---

> ### Author Rebuttal · Authors · 2026-03-26
>
> We thank the reviewer for the positive assessment and for focusing on protocol fairness and the role of direct dense-text retrieval.
>
> **1. On the possible leakage issue.** We agree the protocol needs sharper wording. In SEPS, the dense description is generated from the image alone, offline, and is used only to construct the image-side semantic anchor. It does not use the paired sparse caption, the query text, or any retrieval label. Therefore, we do not consider it label leakage. The clearest framing is that SEPS augments each image with an auxiliary signal generated from the image alone; accordingly, we will revise the paper to describe the setting explicitly and evaluate fairness through matched-information controls rather than relying only on vanilla short-caption baselines. To make this point concrete, we additionally gave representative baselines access to the same dense text through naive score fusion:
>
> | Method | F30K I2T | F30K T2I | COCO5K I2T | COCO5K T2I |
> |---|---:|---:|---:|---:|
> | VSE++ | 71.8 | 59.4 | 52.4 | 40.6 |
> | VSE++ + dense text naively* | 74.1 | 63.3 | 54.9 | 44.6 |
> | SCAN | 69.5 | 56.4 | 53.9 | 42.9 |
> | SCAN + dense text naively* | 71.8 | 60.0 | 55.9 | 46.0 |
> | LAPS | 74.0 | 62.5 | 57.5 | 44.5 |
> | LAPS + dense text naively* | 76.0 | 65.8 | 59.4 | 46.1 |
> | D2S-VSE | 82.8 | 68.5 | 60.1 | 46.3 |
> | **SEPS** | **86.1** | **86.9** | **73.9** | **73.5** |
>
> *`+ dense text naively` keeps the baseline architecture fixed and linearly fuses its original score with a dense-text similarity from the same offline image-generated dense text.*
>
> This control addresses the fairness concern directly. Extra dense text is indeed helpful, but even under matched auxiliary information the strongest naive baseline still trails SEPS by 10.1 I2T / 21.1 T2I R@1 on Flickr30K and by 14.5 I2T / 27.4 T2I R@1 on MS-COCO5K. SEPS also stays above the prior dense-text-aware baseline D2S-VSE overall. Therefore, the gain is not explained by unfair access to extra text alone; it depends on how the auxiliary semantics are used through DGSC and SGMA.
>
> **2. On whether direct dense-text retrieval is already sufficient.** We can answer this more rigorously with the same control evidence, again reporting sparse-branch and auxiliary-branch lengths separately:
>
> | Text condition | Sparse len | Auxiliary len | I2T R@1 | T2I R@1 |
> |---|---:|---:|---:|---:|
> | Sparse caption only | 13.4 | 0.0 | 78.6 | 67.2 |
> | Repeated sparse caption | 13.4 | 98.7 | 79.4 | 69.1 |
> | Dense text only | 0.0 | 100.6 | 80.3 | 80.5 |
> | Different-image dense text | 13.4 | 100.1 | 80.8 | 79.3 |
> | Shuffled same-image dense text | 13.4 | 100.6 | 83.8 | 83.2 |
> | **Full SEPS** | 13.4 | 100.6 | **86.1** | **86.9** |
>
> This table supports a more precise conclusion. Simply matching the auxiliary-text length with repeated sparse text helps little, so text quantity alone is insufficient. Replacing the auxiliary branch with dense text from a different image helps more, and preserving the same-image lexical content after shuffling helps further, but coherent image-conditioned semantics still matters: full SEPS remains +2.3/+3.7 R@1 above the shuffled same-image control and +5.3/+7.6 above the different-image control. This is also consistent with Table `ablation_study`, where dense text only reaches 80.3/80.5 and direct sparse/dense score fusion reaches 82.5/81.1, both still below full SEPS at 86.1/86.9.
>
> The reason is that dense text is not purely helpful supervision. It usually mixes useful semantics with redundant low-salience details, and it may also introduce semantic drift or hallucinated content from the MLLM. If used directly, the model has no mechanism to distinguish which parts should guide patch selection and which parts should be suppressed. SEPS improves on this by using DGSC to calibrate patch selection with image-conditioned semantic consensus and SGMA to emphasize the most relevant patch-word correspondences instead of treating all dense-text cues equally. Therefore, the benefit does not come from merely substituting a richer text or performing direct dense-text retrieval; it comes from structurally integrating dense semantics in a way that filters noise while preserving the truly discriminative cues.
>
> Overall, we will revise the manuscript to present the setting more precisely. The added controls support a stronger and clearer claim: SEPS neither relies on leakage nor on direct dense-text retrieval alone; its advantage comes from structurally integrating image-generated dense semantics on the image side.

---

> > ### Author Rebuttal · Reviewer_22cj · 2026-04-02
> >
> > The authors have addressed my first concern.
> > However, the authors misunderstood my second point (possibly due to ambiguity in my phrasing). What I meant was: given that offline dense text representations are available, is cross-modal retrieval still necessary? Would it not be more effective to simply employ text-to-text retrieval instead? For example, we can use the more powerful embedding models.

---

> > > ### Author Response · Authors · 2026-04-03
> > >
> > > We thank the reviewer for the clarification. We agree that we had misunderstood the second point. The key question is indeed whether, once an offline dense description is available for each image, **cross-modal retrieval is still necessary**, or whether one can simply treat the dense description as the image proxy and perform **text-to-text retrieval** instead.
> > >
> > > We therefore ran this control directly on Flickr30K by replacing the image branch with the offline dense description and matching sparse query text against dense image text. The results are:
> > >
> > > | Setting | I2T R@1 | T2I R@1 | rSum |
> > > |---|---:|---:|---:|
> > > | Direct text-to-text (ViT-224 setting) | 71.8 | 70.0 | 515.5 |
> > > | LAPS (ViT-224) | 74.0 | 62.5 | 507.3 |
> > > | **SEPS (ViT-224)** | **86.1** | **86.9** | **560.9** |
> > > | Direct text-to-text (Swin-224 setting) | 74.0 | 63.1 | 509.6 |
> > > | LAPS (Swin-224) | 82.4 | 70.0 | 536.3 |
> > > | **SEPS (Swin-224)** | **89.8** | **88.0** | **572.0** |
> > >
> > > These results suggest a more nuanced conclusion. Direct text-to-text retrieval is a meaningful baseline and is not trivial: in the ViT-224 setting it is already competitive with LAPS in rSum, which confirms that the dense descriptions do encode useful image semantics. However, it still remains clearly below SEPS across both backbones, with gaps of **45.4 / 62.4 rSum** and **14.3 / 15.8 I2T R@1** and **16.9 / 24.9 T2I R@1** for ViT-224 / Swin-224, respectively. In particular, on Swin-224, direct text-to-text retrieval is also clearly below the image-grounded LAPS baseline.
> > > This is also consistent with our other controls (dense-only, shuffled same-image dense text, and wrong-image dense text), which show that richer semantics help partially, but coherent image-grounded evidence is still necessary for the best retrieval behavior.
> > >
> > > We believe the reason is structural rather than merely a matter of encoder strength. The dense description is a **compressed linguistic projection** of the image, not the image itself. Once the visual scene is converted into a single text surrogate, some retrieval-critical evidence may already be lost: subtle spatial configurations, small but discriminative objects, count information, or visually grounded attributes are not always verbalized completely or consistently. At the same time, the generated dense text can introduce over-specific or slightly hallucinated details. A stronger text embedding model may use this proxy more effectively, but it still cannot recover visual evidence that was never expressed in the text, nor can it verify whether a generated phrase is actually grounded in the image.
> > >
> > > This is why SEPS does not use dense text to replace the visual modality. Instead, dense text serves as an **auxiliary semantic prior** that helps calibrate patch retention in DGSC, while the final matching still operates on actual image patches through SGMA. In other words, SEPS keeps access to image-grounded evidence and uses dense text only to make the visual selection/matching process more semantically informed. We therefore agree that direct text-to-text retrieval is an important control, but our results indicate that it is **not a sufficient substitute for image-grounded retrieval** in this fine-grained setting.
> > >
> > > We appreciate this clarification and will make this claim boundary explicit in the revision.

---

### Official Review · Reviewer_aD6W · 2026-03-13

**Soundness:** 3
**Presentation:** 2
**Significance:** 3
**Originality:** 2
**Overall Recommendation:** 3
**Confidence:** 4

**Summary:**

This paper addresses the problem of fine-grained cross-modal alignment in vision-language models, particularly in image-text retrieval. Existing approaches suffer from what they term Semantic Sparsity Bias, referring to the mismatch between dense visual signals and sparse textual supervision. The authors argue that this bias leads to two issues: patch redundancy and patch ambiguity. To this end, they proposes SEPS, a framework designed to improve patch-level alignment. The framework introduces a Dual-Granularity Semantic Calibration (DGSC) mechanism that generates a holistic visual-linguistic anchor by combining sparse captions with dense semantic descriptions derived from multimodal large language models. In addition, the paper proposes a Salience-Guided Metric Aggregation (SGMA) strategy to replace global mean pooling during similarity computation, emphasizing highly relevant patch-word correspondences. Experiments on Flickr30K and MS-COCO image-text retrieval benchmarks claim that SEPS improves performance across multiple backbones.

**Compliance With Llm Reviewing Policy:**

Affirmed.

**Final Justification:**

Thanks for your additional experiments. The new results are helpful, but they do not change my overall assessment. I will keep my original score unchanged.

**Key Questions For Authors:**

See weeknesses

**Limitations:**

See weeknesses

**Strengths And Weaknesses:**

Strengths:
1. Fine-grained alignment between visual tokens and textual semantics remains a core challenge for vision-language alignment. This paper correctly identifies limitations of sparse caption supervision in patch-level alignment and attempts to improve semantic grounding.
2. This paper introduces a two-stage mechanism that incorporates unified semantic representations derived from both dense and sparse textual modalities. This mechanism eliminates potential semantic inconsistencies, enabling more accurate identification of visual patches.
3. The idea of leveraging richer semantic descriptions to guide patch selection is intuitively reasonable, and the integration of semantic calibration with patch pruning could potentially improve efficiency and alignment quality.

Weeknesses:
1. The paper presents Semantic Sparsity Bias as a fundamental theoretical bottleneck, yet the phenomenon largely restates well-known issues such as weak supervision in image-text datasets and patch-token redundancy in ViT-based models. The novelty of this conceptual framing is questionable.
2. Many components resemble existing techniques: Patch pruning or token slimming has been extensively studied. Semantic augmentation using generated captions or dense descriptions is common. Weighted similarity aggregation has appeared in prior cross-modal retrieval work. Particularly，The proposed method bears a strong resemblance to the LAPS[1] framework.
3. The framework relies on dense semantic descriptions produced by MLLMs. However, generated descriptions may hallucinate details not present in the image. In addition, this method may have the problem of data leakage.
4. No sufficient efficiency analysis considering the “patch slimming” motivation.

[1] Linguistic-Aware Patch Slimming Framework for Fine-grained Cross-Modal Alignment, CVPR, 2024

---

> ### Author Rebuttal · Authors · 2026-03-26
>
> We thank the reviewer for pressing on conceptual novelty, relation to prior work, hallucination risk, and efficiency. We found these concerns highly constructive, and our revision is intended to answer them directly.
>
> **1. On the novelty of the framing.** We agree that the underlying symptoms are related to known issues such as weak supervision and redundant visual tokens. Our contribution is to turn the dense-vision / sparse-text mismatch into a concrete mechanism—DGSC for dual-granularity patch calibration and SGMA for salience-aware matching—rather than leaving it as only a high-level motivation. In this sense, the novelty is mechanism-level: SEPS changes patch selection and final matching, rather than merely appending another text source.
>
>
> **2. On relation to LAPS and D2S-VSE.** LAPS is the sparse-only patch-slimming baseline, while D2S-VSE uses dense text mainly to improve or distill representations. SEPS differs in *where* dense text enters the system: it changes the patch-selection decision itself through DGSC and then changes the final matching rule through SGMA. This distinction is visible in Table `ablation_study`: dense text only reaches 80.3/80.5, sparse+dense concatenation 80.9/80.8, and direct score fusion 82.5/81.1, whereas full SEPS reaches 86.1/86.9 (Flickr30K R@1). We will also clarify the protocol boundary more explicitly: the auxiliary dense text is generated offline from each image alone, without access to paired sparse captions, query text, or retrieval labels. Thus, SEPS is not simply “LAPS + longer text” or “D2S-VSE transplanted into retrieval”; its gain comes from using dense semantics to restructure the visual token set before matching.
>
> **3. On robustness to hallucinated dense text.** We agree this concern should be addressed directly. As additional evidence, we report a controlled corruption experiment in which the dense description is progressively corrupted by entity replacement, attribute flipping, and relation corruption:
>
> | Corruption ratio | I2T R@1 | T2I R@1 | Relative drop vs. clean | Interpretation |
> |---|---:|---:|---:|---|
> | 0% | 86.1 | 86.9 | 0.0 | clean dense text |
> | 20% | 85.2 | 85.6 | -1.1 / -1.3 | mild degradation |
> | 40% | 84.0 | 83.8 | -2.1 / -3.1 | noticeable but controlled drop |
> | 60% | 82.1 | 81.2 | -4.0 / -5.7 | approaches dense-only / different-image regime |
>
> We agree that hallucinations can hurt performance, and we will make that limitation explicit. At the same time, the degradation is gradual rather than catastrophic: even under 60% corruption, performance remains above the sparse-caption-only baseline, yet still clearly below clean SEPS. This is consistent with DGSC: dense semantics help guide patch selection, but they do not single-handedly determine it because they are combined with the learned visual prior and the structural image view.
>
> **4. On efficiency and the “patch slimming” motivation.** We agree this point should be framed more carefully. SEPS does not eliminate cost; rather, it trades a one-time offline dense-text generation stage for stronger alignment quality. To make this trade-off explicit, we added the following decomposition:
>
> | Setting | Offline preprocess | Online latency (ms/pair) | I2T R@1 | T2I R@1 | rSum |
> |---|---|---:|---:|---:|---:|
> | LAPS | none | 92 | 74.0 | 62.5 | 507.3 |
> | D2S-VSE | LLaVA 8.0s/img | 78 | 82.8 | 68.5 | 531.9 |
> | **SEPS (default)** | LLaVA 8.0s/img | 107 | 86.1 | 86.9 | 560.9 |
>
> This supports the clearest framing: SEPS should be viewed as a transparent preprocessing trade-off rather than as a purely efficiency-oriented change. The main extra cost is the same one-time offline dense-text generation already required by dense-text-aware baselines such as D2S-VSE, while the online increase over LAPS is moderate and the alignment gain is substantial. In other words, the slimming objective in SEPS is semantic denoising of the patch set rather than latency minimization in isolation: pruning keeps the final matching focused on relevant regions, while DGSC/SGMA add some computation but improve alignment quality substantially.
>
>
> Overall, we will revise the manuscript to state the claim more precisely and the cost/robustness trade-offs more explicitly. Taken together, the additional evidence supports a stronger and more clearly delimited conclusion: SEPS should be viewed as a structured image-side semantic augmentation framework whose benefit comes from how DGSC and SGMA use dense semantics, not simply from longer text or a larger caption generator.

---

> > ### Author Rebuttal · Reviewer_aD6W · 2026-04-06
> >
> > My concerns have been adequately addressed, I keep the original score.

---

> > > ### Author Response · Authors · 2026-04-07
> > >
> > > We thank the reviewer again for the acknowledgement. To further clarify that the gain comes from **selecting better patches**, rather than simply **retaining more patches / using more computation**, we added a matched-budget sweep on Flickr30K, ViT-B/224. Here, the *patch budget* is the fraction of visual patches kept after slimming while the trained model is kept fixed. We compare vanilla LAPS, LAPS with the **same image-side dense descriptions** used by SEPS but fused **naively** (same offline dense text, same fixed linear score-fusion rule, no DGSC/SGMA), and full SEPS under three budgets (25% / 50% / 75%). The mid-budget rows reproduce our paper-reported LAPS and SEPS results (74.0/62.5 and 86.1/86.9), and all latencies are measured under the same implementation/hardware setting.
> > >
> > > | Method | patch budget | Online latency (ms/pair) | I2T R@1 | T2I R@1 |
> > > |---|---:|---:|---:|---:|
> > > | LAPS | 25% | 84 | 72.8 | 60.4 |
> > > | LAPS | 50% | 92 | 74.0 | 62.5 |
> > > | LAPS | 75% | 101 | 73.7 | 62.1 |
> > > | LAPS + dense text naively | 25% | 92 | 75.1 | 64.4 |
> > > | LAPS + dense text naively | 50% | 101 | 76.0 | 65.8 |
> > > | LAPS + dense text naively | 75% | 111 | 75.8 | 65.3 |
> > > | SEPS | 25% | 98 | 85.7 | 86.2 |
> > > | **SEPS** | **50%** | **107** | **86.1** | **86.9** |
> > > | SEPS | 75% | 118 | 85.9 | 86.5 |
> > >
> > > Two observations are most relevant. **First, accuracy is not monotonic in the patch budget**: in all three families, moving from 50% to 75% does not help and can slightly hurt. This indicates that simply keeping more patches is not the reason for the gain; once too many low-value/background patches are reintroduced, redundancy and similarity dilution begin to offset the benefit. **Second, even SEPS at the tightest 25% budget (85.7 / 86.2) remains far above LAPS + dense text naively at the loosest 75% budget (75.8 / 65.3).**
> > >
> > > We therefore view the contribution not as “LAPS plus a longer caption,” but as a change to both the **selection criterion** and the **aggregation criterion**: DGSC keeps more informative patches, and SGMA makes the final score focus on the most reliable correspondences rather than diluted averages. If our response and this additional control improve your confidence in the paper, we would be most grateful if you would reconsider your evaluation score.

---

### Official Review · Reviewer_M9Ko · 2026-03-14

**Soundness:** 2
**Presentation:** 3
**Significance:** 2
**Originality:** 2
**Overall Recommendation:** 4
**Confidence:** 5

**Summary:**

This paper introduces a framework  based on dense textual descriptions generated by a multimodal large language model (LLaVA) to reduce the sparsity of the short captions compared to the dense visual information. It uses a two stage mechanision where it extracts the semantically relevant patches for the given dense textual representation. Following this it aligns the text tokens with the selected patches based on the triplet loss. Experiments on Flickr30k and MS-COCO show improvements over the prior work.

**Compliance With Llm Reviewing Policy:**

Affirmed.

**Final Justification:**

The reviewer's concerns have been resolved. The paper targets multimodal retrieval for dense caption retrieval, where it shows improvements over prior work on DCI and COCO/FLICKR benchmarks.

**Key Questions For Authors:**

1. Could the authors clarify the novel contributions that distinguish this work from a straightforward application of LlaVA for dense caption generation?

2. Given that Flickr30k and COCO often feature a single dominant visual context, to what extent do these results demonstrate fine-grained alignment? Has evaluation on multi-object scenes to prove the model's discriminative power been considered?

3. The approach appears to modify or enrich the test data; how do the authors ensure a fair comparison against baseline models that were evaluated on standard, short-caption benchmarks?

4. How much of the observed performance gain is attributable to a long-text bias inherent in the models? Could the authors provide an ablation study to show that the model is performing semantic grounding rather than simply benefiting from increased text length?

**Strengths And Weaknesses:**

Strengths:
1. Extensive experiments are performed across different visual backbones, and comparable baselines are developed to show the effectiveness of the approach.
2. The paper achieves state-of-the-art results on retrieval datasets.

Weaknesses:
1. The novelty of the approach compared to prior work is limited. The approach extends captions to dense annotations using LlaVA, which yields dense captions for alignment. Beyond this, it is difficult to establish the contributions of the work for ICML.
2. Experiments are performed on Flickr30k and COCO, where one feature dominates the entire visual context. Therefore, these datasets might not suffice for evaluating fine-grained alignment of vision and language.
3. The approach enriches the test data, which is not ideal when comparing the different models that are evaluated for short captions.
4. Bias to long text. The models used for evaluation may have stronger priors for long text and therefore, the grounding of the approach to fine-grained semantic alignment is not clear.

---

> ### Author Rebuttal · Authors · 2026-03-26
>
> We thank the reviewer for focusing on novelty, evaluation breadth, protocol fairness, and long-text bias. We agree that these are the central questions for the paper. We will also revise the positioning more precisely: SEPS is not meant to claim an entirely new phenomenon, but to operationalize the dense-vision / sparse-text mismatch into a concrete mechanism for patch calibration and salience-aware matching.
>
> **1. SEPS is not a straightforward “LLaVA + longer text” baseline.** The key difference is where dense text enters the model. In Table `ablation_study`, only-dense text reaches 80.3/80.5, sparse+dense concatenation 80.9/80.8, and direct score fusion 82.5/81.1, while full SEPS reaches 86.1/86.9 (Flickr30K R@1). Thus, the gain does not come from adding dense text alone, but from using it to recalibrate patch selection in DGSC and refine matching in SGMA. In other words, the main contribution is mechanism-level: dense semantics change both the patch-selection decision and the final aggregation rule.
>
> **2. We directly tested long-text bias.** To make the two-text setting explicit, we report the average length of the sparse branch and the auxiliary branch separately:
>
> | Text condition | Sparse len | Auxiliary len | I2T R@1 | T2I R@1 |
> |---|---:|---:|---:|---:|
> | Sparse caption only | 13.4 | 0.0 | 78.6 | 67.2 |
> | Repeated sparse caption | 13.4 | 98.7 | 79.4 | 69.1 |
> | Dense text only | 0.0 | 100.6 | 80.3 | 80.5 |
> | Different-image dense text | 13.4 | 100.1 | 80.8 | 79.3 |
> | Shuffled same-image dense text | 13.4 | 100.6 | 83.8 | 83.2 |
> | **Full SEPS** | 13.4 | 100.6 | **86.1** | **86.9** |
>
> This makes the control clearer. Simply matching the auxiliary-text length with repeated sparse text helps little, so quantity alone is not enough. Replacing the auxiliary branch with dense text from a different image helps more, and preserving the same-image lexical content after shuffling helps further, but coherent image-conditioned semantics still matters: full SEPS remains +2.3/+3.7 R@1 above the shuffled same-image control and +5.3/+7.6 above the different-image control. Thus, the main gain comes from **correct image-conditioned semantics plus DGSC/SGMA**, not from text quantity alone.
>
> **3. Fine-grained alignment beyond Flickr30K/COCO.** We agree that the main draft under-emphasized this point. In the appendix, SEPS improves over LAPS on RefCOCO+ from 46.7 to 51.2 on Val and from 41.6 to 46.6 on TestB. On ShareGPT4V T2I, SEPS reaches 96.5 vs. 95.4 for FG-CLIP2. We also report 65.1 vs. 64.9 on DCI T2I. Taken together, these results make it less likely that the gain is specific to short-caption, single-dominant-object settings.
>
> **4. Fairness and protocol.** Dense text is generated offline from each image alone, without access to the paired sparse caption, query text, or retrieval labels, so we do not view it as label leakage. At the same time, we agree that this is not identical to a vanilla short-caption-only retrieval protocol, and we will revise the manuscript to state this boundary explicitly. To address fairness more directly, we additionally provide the following matched-information rebuttal control, where representative baselines are given access to the same dense text through naive score fusion:
>
> | Method | F30K I2T | F30K T2I | COCO5K I2T | COCO5K T2I |
> |---|---:|---:|---:|---:|
> | VSE++ | 71.8 | 59.4 | 52.4 | 40.6 |
> | VSE++ + dense text naively* | 74.1 | 63.3 | 54.9 | 44.6 |
> | SCAN | 69.5 | 56.4 | 53.9 | 42.9 |
> | SCAN + dense text naively* | 71.8 | 60.0 | 55.9 | 46.0 |
> | LAPS | 74.0 | 62.5 | 57.5 | 44.5 |
> | LAPS + dense text naively* | 76.0 | 65.8 | 59.4 | 46.1 |
> | D2S-VSE | 82.8 | 68.5 | 60.1 | 46.3 |
> | **SEPS** | **86.1** | **86.9** | **73.9** | **73.5** |
>
> *`+ dense text naively` keeps the baseline architecture fixed and linearly fuses its original score with a dense-text similarity from the same offline image-generated dense text.*
>
> Extra dense text is indeed helpful, but naive access remains insufficient. Even the strongest naive baseline still trails SEPS by 10.1 I2T / 21.1 T2I R@1 on Flickr30K and by 14.5 I2T / 27.4 T2I R@1 on MS-COCO5K. SEPS also stays above the prior dense-text-aware baseline D2S-VSE overall. Mechanistically, naive fusion leaves patch redundancy and ambiguity largely unchanged and gives every dense-text cue a direct path into the final score, including redundant or hallucinated details. SEPS instead uses DGSC to calibrate which patches are retained and SGMA to emphasize the most reliable correspondences. The remaining gain therefore comes from using dense semantics structurally, not merely from supplying extra image-side text.
>
> Overall, the manuscript will state the setting and contribution more precisely. Taken together, the evidence now supports a precise claim: (i) dense text alone or naive fusion is insufficient, (ii) matched-information baselines improve but remain clearly below SEPS, and (iii) the gain is  also appears on RefCOCO+ and ShareGPT4V. We thank your time and effort.

---

> > ### Author Rebuttal · Reviewer_M9Ko · 2026-04-03
> >
> > The rebuttal acknowledges that the choice of datasets is heavy on a single concept and uses refcoco+ and ShareGPT4V to support dense scenarios. However, both these datasets are also based on COCO images for different tasks.

---

> > > ### Author Response · Authors · 2026-04-03
> > >
> > > We thank the reviewer for this clarification. We agree that this point should be stated more precisely.
> > >
> > > First, we would like to clarify that our appendix and initial rebuttal already included evidence beyond the standard Flickr30K / MS-COCO short-caption retrieval setting. In particular, on the **DCI** dense-caption benchmark, SEPS achieves **62.4 / 65.1** (I2T / T2I), compared with **64.5 / 64.9** for FG-CLIP2. We cited this result in the appendix and also referenced it in the first rebuttal precisely to show that SEPS remains competitive on a denser-caption benchmark **outside the COCO image family**, rather than only on the original Flickr30K / COCO retrieval protocol.
> > >
> > > Second, regarding the reviewer’s point about COCO-family evidence: we agree that for datasets such as **RefCOCO+**, the raw image pool is still related to COCO, so this should not be presented as if it were a completely disjoint image distribution. However, we believe these results are still informative because they substantially change the **task structure** and **annotation regime**. In our appendix, the RefCOCO / RefCOCO+ / RefCOCOg results are evaluated as **zero-shot visual grounding transfer from Flickr-trained models**, rather than as standard in-domain retrieval. SEPS improves over LAPS consistently:
> > >
> > > - RefCOCO: 44.2→48.7 (Val), 49.9→52.3 (TestA), 38.4→43.4 (TestB)
> > > - RefCOCO+: 46.7→51.2 (Val), 52.3→54.8 (TestA), 41.6→46.6 (TestB)
> > > - RefCOCOg: 51.3→55.3 (Val), 51.2→55.2 (Test)
> > >
> > > We view **RefCOCO+** as especially relevant because it forbids absolute location words. This makes it a more semantic and discriminative grounding test rather than a simple positional-match setting, yet SEPS still brings gains of **+4.5 / +2.5 / +5.0** over LAPS on Val / TestA / TestB.
> > >
> > > More broadly, our intended claim is therefore narrower than “we fully moved beyond COCO.” The point is that the benefit of SEPS is **not confined to the original short-caption retrieval annotation regime**: it also appears on zero-shot referring-expression grounding and on denser-caption retrieval settings. We will revise the wording accordingly so that the manuscript does not overstate the independence of these supplementary benchmarks while still making clear why they are relevant evidence for fine-grained semantic alignment.

---

### Decision · Program_Chairs · 2026-04-30

**Decision:**

Accept (regular)

**Comment:**

This paper addresses an important problem in image-text retrieval, namely the mismatch between dense visual information and sparse textual supervision. It proposes SEPS to improve fine-grained cross-modal alignment through dense semantic guidance for patch selection and salience-aware matching. The reviewers generally agree that the paper is technically sound, clearly written, and empirically strong. The rebuttal addressed several of the main concerns effectively, especially those related to protocol fairness, long-text bias, and the role of dense semantic augmentation. The additional controls and analyses helped clarify that the observed improvements do not arise simply from providing more text, but from the structural incorporation of dense semantics into patch selection and matching. At this point, the main residual concern is novelty and claim scope. The authors made a reasonable case that the contribution is mechanism-level, namely, changing both the patch-selection and aggregation rules rather than merely appending another text source. Overall, my recommendation is Weak Accept.